Sample entropy analysis for the estimating depth of anaesthesia through human EEG signal at different levels of unconsciousness during surgeries

Liu Quan 1
Ma Li 1
Fan Shou-Zen 2
Abbod Maysam F. 3
Shieh Jiann-Shing jsshieh@saturn.yzu.edu.tw 4
1 School of Information Engineering, Wuhan University of Technology , Wuhan , China
2 National Taiwan University , Tapei , Taiwan
3 Department of Electronic and Computer Engineering, Brunel University London , Uxbridge , United Kingdom
4 Department of Mechanical Engineering and Innovation Center for Big Data and Digital Convergence, Yuan Ze University , Taoyuan , Taiwan
Gomez Shawn
Electronic publication date: 2018 May 23
Publication date: 2018
Volume: 6
Electronic Location ID: e4817
Received 2017 Nov 15; Accepted 2018 May 1
Copyright: ©2018 Liu et al.
Copyright year: 2018
Copyright holder: Liu et al.
License: This is an open access article distributed under the terms of the Creative Commons Attribution License, which permits unrestricted use, distribution, reproduction and adaptation in any medium and for any purpose provided that it is properly attributed. For attribution, the original author(s), title, publication source (PeerJ) and either DOI or URL of the article must be cited.
License URL: https://creativecommons.org/licenses/by/4.0/

Keywords: Electroencephalogram, Sample entropy, ANOVA, Depth of anaesthesia, Random Forest

Funding: Innovation Center for Big Data and Digital Convergence, Yuan Ze University National Chung-Shan Institute of Science & Technology in Taiwan XB06183P478PE-CS Wuhan University of Technology 2015-JL-012 National Natural Science Foundation of China 51475342 51675389 This work was supported by the Innovation Center for Big Data and Digital Convergence, Yuan Ze University, Taiwan, which are sponsored by Ministry of Education, and the National Chung-Shan Institute of Science & Technology in Taiwan. It is also supported by the Wuhan University of Technology international exchange program (No. 2015-JL-012) and the National Natural Science Foundation of China (No. 51475342,51675389). The funders had no role in study design, data collection and analysis, decision to publish, or preparation of the manuscript.

==============================
Estimating the depth of anaesthesia (DoA) in operations has always been a challenging issue due to the underlying complexity of the brain mechanisms. Electroencephalogram (EEG) signals are undoubtedly the most widely used signals for measuring DoA. In this paper, a novel EEG-based index is proposed to evaluate DoA for 24 patients receiving general anaesthesia with different levels of unconsciousness. Sample Entropy (SampEn) algorithm was utilised in order to acquire the chaotic features of the signals. After calculating the SampEn from the EEG signals, Random Forest was utilised for developing learning regression models with Bispectral index (BIS) as the target. Correlation coefficient, mean absolute error, and area under the curve (AUC) were used to verify the perioperative performance of the proposed method. Validation comparisons with typical nonstationary signal analysis methods (i.e., recurrence analysis and permutation entropy) and regression methods (i.e., neural network and support vector machine) were conducted. To further verify the accuracy and validity of the proposed methodology, the data is divided into four unconsciousness-level groups on the basis of BIS levels. Subsequently, analysis of variance (ANOVA) was applied to the corresponding index (i.e., regression output). Results indicate that the correlation coefficient improved to 0.72 ± 0.09 after filtering and to 0.90 ± 0.05 after regression from the initial values of 0.51 ± 0.17. Similarly, the final mean absolute error dramatically declined to 5.22 ± 2.12. In addition, the ultimate AUC increased to 0.98 ± 0.02, and the ANOVA analysis indicates that each of the four groups of different anaesthetic levels demonstrated significant difference from the nearest levels. Furthermore, the Random Forest output was extensively linear in relation to BIS, thus with better DoA prediction accuracy. In conclusion, the proposed method provides a concrete basis for monitoring patients’ anaesthetic level during surgeries.

Introduction

General anaesthesia (GA) is a temporary drug-induced unconsciousness state, which is reversible under human manipulation (Purdon et al., 2015). Modern GA is widely used in clinical operations because it guarantees safe surgical operations for millions of patients every year (Vutskits & Xie, 2016). However, numerous published articles have provided strong evidence of postoperative side effects, such as delirium (Lepouse et al., 2006) and cognitive dysfunction (Monk et al., 2008). Limited solutions are available to deal with these problems effectively because the underlying mechanism of anaesthesia is still not entirely understood (Uhrig, Dehaene & Jarraya, 2014). Therefore, optimal titration of anaesthetic dose is crucial to achieve the ideally effective analgesia, unconsciousness, and immobility for reducing the potential negative influences of either underdosing or overdosing. Thus, a precise evaluation of the depth of anaesthesia (DoA) is crucial due to the emerging critical requirements of patients’ surgery safety and experience.

Anaesthetic agents mainly rely on enhancing the activity of inhibitory cells or suppressing the activity of excitatory cells. Gamma-aminobutyric acid (GABA) and N-methyl-D-aspartate (NMDA), pervasive receptors, are usually induced by drugs including propofol, sevoflurane, and ketamine (Brown, Lydic & Schiff, 2010; Uhrig, Dehaene & Jarraya, 2014). In general, some disruption of neural communications occurs during the loss of consciousness (LoC). Anaesthetics alter the brain’s normal information processing. Thus, discovery of how to assess the degree of the brain state might reveal a feasible method for determining DoA; researchers have regarded this line of inquiry as promising (Shalbaf et al., 2013). Nevertheless, the monitoring of the brain state is not routinely preferred in practical anaesthesia care because markers that reliably reflect consciousness level variations under GA have yet to be identified (Palanca, Mashour & Avidan, 2009). The current standards for indirect DoA measurements, by which anaesthesiologists judge whether patients are adequately anaesthetised, include: (1) observing changes in the heart rate, blood pressure, and muscle tone and (2) monitoring the drug pharmacokinetics, pharmacodynamics, and the level of exhaled anaesthetic gas of the inhaled anaesthetics (Purdon et al., 2013). Anaesthesiologists primarily rely on autonomic and behavioral responses to optimize levels of anaesthesia and provide appropriate analgesia. However, individual anaesthesiologists apply these traditional methods idiosyncratically. Moreover, analysis of these signs can be difficult and unreliable due to the use of some medications such as muscle relaxants (Shalbaf et al., 2013). Accumulating evidence shows that electrical potential in the frontal cortex, namely electroencephalogram (EEG), is highly associated with GA (Kiersey, Bickford & Faulcone Jr, 1951; Martin, Faulconer & Bickford, 1959). Brain information processing is based on transmission and reception of the spike or action potential by neurons, which is one of the fundamental mechanisms underlying central nervous system interactions (functions) (Kandel et al., 2000). Information processing is altered by drugs (Vizuete et al., 2014). They accordingly change neural behaviour, such as the rate of firing, which can definitely be reflected by the neurons’ electrical potential. However, subcortical areas produce weak and difficult-to-detect electrical potentials because the electric field strength decreases as the distance of a potential from its source increases (Hämäläinen et al., 1993). Scalp EEG patterns reflect the states of both cortical and subcortical structures due to their strong connections (Ching et al., 2010). Thus, EEG can be used to explore the interactions between different structures under anaesthetics. EEG behaviour has been proved to be strongly correlated with different DoA levels (Brown, Lydic & Schiff, 2010; Purdon et al., 2013). One typical phenomenon is the burst-suppression pattern. Therefore, the EEG is the most pervasive noninvasive signal that reflects the neural communications and states. Therefore, numerous EEG-based DoA-level-monitoring machines have been produced and are described in following passages.

Fortunately, various electrophysiological measurements of brain activity have been investigated to quantify DoA (McKeever, Johnston & Davidson, 2014; Poorun et al., 2016). Spectrum-power analysis involves the evaluation of specific frequency component decompositions, compressed spectral arrays, two-dimensional plots of the spectrogram, and so on (Billard et al., 1997). Moreover, several spectral indices have been proposed, such as the 95% spectral edge frequency (SEF) (Katoh, Suzuki & Ikeda, 1998), median frequency (MF) and spectral entropy (SE) (Höcker et al., 2010). Among them, the SE has already been applied to the commercial monitor Datex–Ohmeda S/5 (GE Healthcare, Helsinki, Finland). However, these indices are used for studying spectral characteristics which are based on traditional fast Fourier transform (FFT) techniques. Raw EEG data spectrogram is often investigated to identify the difference between the awake state and unconsciousness state (Akeju et al., 2014; Purdon et al., 2013) for overcoming some drawbacks of index currently applied though they just compared the unique features of spectrogram under different conditions. However, it is difficult to practically distinguish the states. Most frequency analysis methods, which are based on Fourier theory, assume the data is stationary in a short period. Because an EEG presents non-stationary and nonlinear features in human biological system, those frequency based methods may ignore valuable information. Consequently, it is reasonable to apply the nonlinear dynamics and information theory to EEG for the prediction of DoA. Besides, various other studies only focus on the segment comparisons (Akeju et al., 2014). Therefore, it is crucial to conduct the whole perioperative studies. Moreover, few researchers use other physiological methods such as electrocardiography (ECG) (Liu et al., 2017b) or blood pressure combined with the heart rate (Shalbaf, Behnam & Moghadam, 2015). However, it is obvious that anaesthetics affect the brain more directly, which results in the variations of EEG signals.

In the past decade, the concept of entropy has been widely utilised in medical and neurological techniques such as the approximate entropy (ApEn) (Bruhn et al., 2001) and Shannon entropy (Bruhn et al., 2001; Shannon, 2001). The concept of entropy is derived from the time series. Sample Entropy (SampEn), an improved form of the previous entropy methods (Richman & Moorman, 2000), estimates the probability that data series are closely correlated in a dataset within a given tolerance. SampEn calculates the irregularity of a time series and reflects the randomness and complexity in time domain. In general, regular systems have lower values of entropy and vice versa. In particular, SampEn applies to non-stationary data and is more noise resistant, which overcomes the drawbacks of many spectral methods that are mostly based on linear Fourier transform. Therefore, SampEn has been applied to many biomedical dynamics such as heart rate variability and EEG analysis (Chu et al., 2017; Huang et al., 2013; Shalbaf et al., 2012). Previous research efforts, including ours, ignored the variability in different anaesthetic levels to some extent. Unconsciousness is not only an unresponsive state but also is notably associated with the concentration level of anaesthetic agents, which affect both neuronal activity and brain networks. Moreover, any disruption in the neuronal activity and brain networks may lead to conditions such as burst suppression or postoperative cognition dysfunction (Purdon et al., 2015). These adverse effects may be avoided if DoA can be precisely controlled using an appropriate drug dosage. Therefore, further studies should be conducted for assessing DoA at different levels.

Currently, many machine learning methods receive much attention for classification and regression in data mining, such as Support Vector Machine (SVM), Artificial Neural Network (ANN) and Random Forest. SVM, a type of supervised learning method, depends on one or multiple hyperplanes to be used in classification and regression (Drucker et al., 1997). The larger the margin is from the training data to the hyperplane, the better performance is. ANN is an advanced modelling tool in statistics, machine learning, and cognitive science (Alpaydin, 2014; Kriegeskorte, 2015). Neural network models are widely applied to theoretical neuroscience and computational neuroscience. In general ANN implementations, artificial neurons and connections typically adapt their connection weights on the basis of a learning procedure that maps input signals to outputs. An ANN is organized in layers that exhibit different types of transformations on their inputs. Random Forest is an ensemble learning method and can be considered as an ensemble of decision trees used for classification, regression and other tasks (Liaw & Wiener, 2002). By a random selection of features at each tree node, the correlation between the trees in the forest can be decreased thus decreasing the forest error rate and reducing over fitting.

In this paper, EEG analysis was conducted to estimate DoA using Bispectral index (BIS) as the reference value (Aspect Medical Systems, Newton, MA, USA). BIS is the most commonly used commercial index, it is considered as the gold standard to evaluate our method performance due to its relatively high accuracy in the GABA receptor anaesthetic situation and the BIS has been approved by the US Food and Drug Administration (FDA). The range of the recorded BIS is from 0 (EEG suppression state) to 100 (awake state). The BIS index has significant advantages such as comprehensive clinical validation (Johansen & Sebel, 2000; Luginbühl et al., 2003). However, the algorithm remains unreleased and has some specific weak points which are comprehensively discussed in section ‘Discussion’, thus posing the need for a superior open source methodology. In this study, the focus is on patients receiving GABA-receptor dependent anaesthetic agents by employing BIS as the gold standard for preliminary investigation due to its precision when using specific anaesthetic drugs. In total, 24 datasets of EEG data were analyzed to evaluate DoA. First, Multivariate Empirical Mode Decomposition (MEMD), a non-linear, non-stationary method, was applied to filter the noise. Subsequently, SampEn features were extracted from specific combinations of Intrinsic Mode Function (IMF), generated from MEMD. To analyse the performance of SampEn, it was compared with recurrence quantification analysis (RQA) measurement and permutation entropy (PEn). Next, three regression models were chosen to train the features for regression for optimising the performance. Finally, the capability of the models was evaluated by comparing the outputs of these models with the BIS gold standard. In order to analyze this method capability in more detail, analysis of variance (ANOVA) analysis was undertaken for the data pairs of four subgroups.

Materials and Methods

Subjects

Ethical approval was granted by the Research Ethics Committee of the National Taiwan University Hospital (NTUH) (No: 201302078RINC). Written informed consent forms were obtained from all the patients who were involved. In this study, patients who received regular surgery under GA were recruited from NTUH (i.e., no patients underwent high risk operations related to brain, heart, lung, etc.). Those who had alcohol or smoking habits, or had other medical illness that affected the data recording, were excluded. Finally, 24 patients (ASA I or II, age (yr): 44.5 ± 12.9, height (cm): 164.2 ± 7.1, weight (kg): 63.4 ± 14.8, and BMI (kg/m2): 23.4 ± 4.2, gender: 14 females/10 males) were eligible for this study. Operations last 126.4 ± 72.9 min from anaesthetic medical recording forms.

Anaesthetic procedure manipulation

The anaesthetic procedures, as shown in Fig. 1, can be illustrated as follows: first, patients’ clinical information, including height, weight, age, gender, and type of operation performed, were provided to anaesthesiologists for preparing clinical assessment of the anaesthetic plan. Prior to the beginning of this study, patients were required to eat nothing for at least 8 h. All the mandatory procedures were conducted such as routine and emergency drug check. Then each patient received an appropriate volume of anaesthetic agents for the regular operations. In this study, intravenous propofol was used to induce unconsciousness along with some other compounds such as muscle relaxants and analgesic. Propofol is a GABA-receptor dependent drug, which provides a rapid anaesthetic effect within a few minutes. When patients lost consciousness (i.e., no response to verbal commands or stimulus), the next stage was attained. For the maintenance period, either propofol was titrated for short-term surgeries or gas anaesthetics (desflurane or sevoflurane) together with air and oxygen in order to maintain unconsciousness. Practically, the concentration levels were adjusted within 1–1.5 minimal alveolar concentration. The anaesthetic gases were delivered by a medical mask. When approaching the end of a surgery, additional drugs are administrated to comfort the patients like vagostin, atropine, morphine, etc. All the anaesthetic agents’ information is detailed in Table 1. To maximize the universality of the data, drugs were not specified. Thus, patients were often injected with a combination of drugs used in ordinary operations. General anaesthesia was performed safely through all stages by monitoring the physiological signals such as EEG, ECG, photoplethysmography (PPG) and by examining the intermittent vital signs of blood pressure (BP), heart rate (HR), pulse rate (PR), pulse oximeter oxygen saturation (SpO2), and so on. These measurements were shown on the physiological monitor MP60 (Intellivue; Philips, Foster City, CA, USA). If any observed signal changed irregularly, doctors will adjusted the intraoperative standard anaesthesia machine accordingly. The surgical procedure and anaesthetic management details were acquired from the anaesthetic recording sheets by the hospital staff.

Figure 1 The anaesthetic procedure conducted in our study.

ASA, American Society of Anaesthesiologists.

Table 1 Properties of drugs administrated in the anaesthetic treatment procedures.

For maintenance section, we show both the concentration level of the inhaled drugs and the time duration. Note that the dosage of maintenance drugs varies practically, thus showing with a clinically common range (not every subject receives all the drugs).

Anaesthetic management	Mean (std)	
Induction combinations	
Propofol (mg)	119.6(25.1), n = 23	
Fentanyl (mg)	116.7(20.9), n = 24	
Atropine (mg)	0.6(0.19), n = 9	
Nimbex (mg)	9.6(2.1), n = 10	
Xylocaine (mg)	42.0(12.3), n = 20	
Maintenance	
Sevoflurane (%, min)	2∼3, 146.8(75.3), n = 6	
Desflurane (%, min)	6∼9, 123.8(80.4), n = 16	
Propofol ( µg, min)	1.5∼4, (126,100)a, n = 2	
Additional drugs administered close to the end	
Morphine (mg)	6.7(2.9), n = 3	
Ketamine (mg)	18.3(7.3), n = 3	
Atropine (mg)	1(0), n = 21	
Vagostin (mg)	2.4(0.2), n = 23	
Notes.

a Only two cases maintained with propofol, therefore, we showed both dosage values respectively.

Data recording

Prior to EEG recording, a conductive paste was used to improve the contact between the frontal scalp skin and EEG BIS™ Quatro Sensor (Aspect Medical Systems, Newton, MA, US) in order to reduce the impedance under 5 kΩ. The EEG sensor was connected to the MP60 machine via the standard factory connection cable. Raw continuous EEG waveform data; including the aforementioned discrete routine vital signals; were recorded on a laptop through the serial RS-232 port utilising software developed in Borland C++ Builder 6 (C++ version 6; Borland Company, Austin, TX, USA). The EEG data were acquired at a sampling rate of 128 Hz. The intermittent BIS value was obtained every 5 s. Other intermittent vital signs such as, HR, PR, BP, and SPO2 were also recorded every 5 s using specific sensors with MP60 (e.g., pulse oximetry, ECG sensor and blood pressure cuff). The recording began 5 min ahead of the onset of induction, at which time patients were fully awake, and it terminated when patients began to respond to doctors either by voice or movement.

Data preprocessing

All the clinical administration and physiological data were sorted in order. Subsequently, tasks such as conversion of the EEG data format and labelling of the event time points were conducted. All cases’ data were then visually inspected roughly to remove the specific segments of artefacts resulting in waveform saturation (e.g., electrical artefacts caused by medical equipment or body movement), thus considerably reducing the outlier point segments. A notch filter was used to remove the 60-Hz line noise. Subsequently, a 5 s data epoch was used to conduct the MEMD analysis, and a 30-s segment of the reconstructed signal which was used for the SampEn calculation. The window step size was set to 5 s to be consistent with the BIS value for further comparisons.

Data analysis

Filtering by MEMD

EEG signal can be easily contaminated by artefacts such as noise from electromyography (EMG), electrooculography (EOG), and electric circuits (Huang et al., 2013). Thus, it is necessary to filter the signals to reduce the computation error. Regarding the nonlinear and nonstationary characteristics of EEG, a method known as the empirical mode decomposition (EMD) was proposed by Huang et al. (1998). The EMD method decomposes signal into a series of IMFs. The extraction procedure of an IMF from a signal is known as sifting. In this study, all the local extrema were found, and then, the upper envelope was generated by drawing a cubic spline line through all the local maxima. Similar steps were conducted for the local minima to produce the lower envelope.

All the data should fall in the range between the upper and lower envelopes. The first component h1 is obtained by computing the difference between the data and mean: (1) Xt−m1=h1

where m1 denotes the mean, and h1 can only be considered as a proto-IMF. In the next step, h1 is treated as data: (2) h1−m11=h11.

After sifting up to k times, h1 becomes an IMF as follows: (3) h1k−1−m1k=h1k.

Then, the first IMF is acquired as h1k: (4) C1=h1k.

One can repeat these steps for the residual generated by previous loops to obtain successive IMFs.

Finally, the signal can be decomposed as: (5) Xt= ∑i=1Ncit+rNt

where N is the number of IMFs, ci(t) is the ith IMF and rN(t) is the residual. A combination of different IMFs can reconstruct a signal to eliminate the artefacts. However, EMD induces the mode mixing problem. An improved EMD known as MEMD was presented by Rehman & Mandic (2009) and noise assisted MEMD (N-A-MEMD) was also introduced by (Ur Rehman & Mandic, 2011). The new N-A-MEMD solves the mode mixing problems, and also can be applied to single channel data by adding Gaussian noise together to constitute multichannel data. The definition is as in Eq. (6). (6) mt=1K∑i=1Keθit

where eθi(t) is the multivariate envelope curves of the whole set of direction vector, K is the length of the vectors, and m(t) is calculated by means of the multivariate envelope curves. Unlike the traditional method that has constant amplitude and frequency in the harmonic component, an IMF provides a simple oscillatory mode as a counterpart and has a variable amplitude and frequency over time. Because decomposition is based on the local characteristics of a data series, it is applicable to nonlinear and nonstationary processes, thus overcoming both pseudo-linearity and stationarity. In our previous study (Huang et al., 2013), the EEG signals were reconstructed by summing IMF2 and IMF3 after decomposition to obtain the filtered signals due to their superior discrimination ability of different states of anaesthesia.

SampEn analysis

Due to the non-stationary characteristic of EEG data, SampEn is appropriate to quantify the degree of irregularity of EEG data. A highly complicated series generates high SampEn values. SampEn, proposed by Richman & Moorman (2000), improves ApEn to more precisely measure the complexity of a physical time series by following steps:

Step 1: Obtain a time series with N points Xi,1≤i≤N. Set parameters r and m, which represent tolerance and embedding dimension, respectively.

Step 2: The m-dimension template vector Xmi is defined as: (7) Xmi=Xi,Xi+1,…,Xi+m−1,1≤i≤N−m+1.

Step 3: Calculate the distance between two vectors: (8) dXmi,Xmj= maxXmi+k−Xmj+k0≤k≤m−1,i≠j.

Step 4: Let Bi be the number of vectors Xmj within r of Xmi, then: (9) Bimr=1N−m+1Bi

(10) Bmr=1N−m∑i=1N−mBimr.

Step 5: If we set m = m + 1 and repeat Step 1 to Step 4, we obtain the following: (11) Amr=1N−m∑i=1N−mAimr.

Finally, SampEn is obtained as: (12) SampEn=−lnAmrBmr.

Before computing the SampEn, three crucial parameters must be set: the length of the time series N, tolerance r, and dimension m. N was chosen to be 3,750 points (30 s). Various theoretical and clinical applications have described the validity of m = 1 or 2 and r in the range of 0.1 × std − 0.2 × std, where std is the standard deviation. The algorithm was initialized with the following parameters: m = 1 and r = 0.1 × std.

In addition, permutation entropy and recurrence analysis are utilised to analyse the nonstationary EEG data. Permutation entropy is an ordinal analysis method and is quantified by dividing time series into ordinal patterns for describing the order of relations between the present values and a fixed number of equidistant past values. Researchers claim it offers simplicity, robustness, and low computational complexity (Zanin et al., 2012); these advantages are mostly required in nonstationary data analysis, including studies related to anaesthesia (Li, Cui & Voss, 2008). In this study, we chose dimension m = 6 and lag τ = 1, as suggested in previous studies (Li, Cui & Voss, 2008; Liu et al., 2016). Recurrence analysis is a kind of statistical analysis method used to quantify the frequency of the phase space trajectory of a dynamical system visiting approximately the same area in some phase space (Huang, Wang & Singare, 2006; Marwan et al., 2007). We used the RQA tool to conduct the analysis (Marwan et al., 2007). The parameters were chosen: embedding dimension equal to 3, delay equal to 5, and threshold equal to 0.5, which were decided by the global false nearest neighbour and mutual information. In this study, both recurrence rate (RR) and determinism (DET) were used to measure the recurrence analysis. The data segment window size and step were the same as in the SampEn processing part.

Regression models

In order to symbolise DoA more accurately, three typical regression models were applied to train our datasetdash SVM, ANN and Random Forest. These models serve vital purposes in machine learning and have strong self-learning capabilities. Libsvm was employed to conduct SVM regression (Chang & Lin, 2011; Drucker et al., 1997). The exponential function was chosen as the kernel. From engineering perspective, three to four layers were mostly used for ANN (Kourentzes, Barrow & Crone, 2014; Ripley & Ripley, 2001). The backpropagation neural network (BPNN), a structure of 1–9–18–1, was chosen as the model after trials. The commonly recognised BIS values were regarded as the target whereas the entropy features were employed as input data. The input sample data was randomly divided into training (70%), validation (15%), and testing datasets (15%) to adjust the weights and bias for generating the eventual model. Random Forest is a type of ensemble of trees. We used 1,000 trees to train the SampEn extracted from filtered EEG. For all the training processes, the corresponding BIS acted as a target as well.

Statistical analysis

To verify the proposed index capability for decimating the anaesthesia stages, parametric paired Student t-test was used. For the comparisons of intact surgery assessment of DoA, the receiver operating characteristic curve (ROC) was calculated to obtain area under the curve (AUC), which is often used when diagnosing diseases. The BIS binary threshold for distinguishing between anaesthesia condition and awake state is set to 65 (Johansen, Sebel & Sigl, 2000). Pearson’s correlation coefficient and mean absolute error (MAE) were computed between the index and BIS value. To pursue the most appropriate regression model, the Pearson correlation coefficient, MAE, and AUC were calculated for each model by comparing output to the gold standard BIS. Moreover, DoA states were classified into four groups based on the BIS (Johansen, Sebel & Sigl, 2000). ANOVA analysis was conducted for the corresponding stage period SampEn to study the performance in detail. The statistical analysis was performed using SPSS Statistics (IBM v22; Armonk, NY, USA) and MATLAB (Mathworks R2014b; Natick, MA, USA).

The entire workflow diagram is shown in Fig. 2. First, the valid subjects were selected, and then, the preprocessing was conducted. Afterwards, MEMD was applied to filter the data, thus constructing new data by combining IMF2 and IMF3. Then, SampEn features were extracted from data segments for all cases followed by ANN modeling. Finally, the results were statistically compared to BIS to evaluate the performance, including the perioperative analysis evaluation and detailed sub-level group ANOVA analysis.

Figure 2 Diagram of the general work flow.

N-A-MEMD, noise assisted multivariate empirical mode decomposition; IMF, instrinsic mode function; BIS, bispectral; DoA, depth of anaesthesia; RoC, receiver operating characteristics; ANOVA, analysis of variance.

Results

MEMD filter performance

As discussed previously, the non-stationary and non-linear characteristics of physiological signals (i.e., EEG in this study) require a robust and appropriate method to eliminate the weakness of traditional FFT methods. MEMD was applied to reconstruct the signal. A representative patient’s SampEn derived by summing both the combination signal (i.e., IMF2 + IMF3) and raw EEG data were presented in Fig. 3 with the gold standard BIS. By a rough visual evaluation, it is obvious that the filtered outcome behaved better than the raw one. Statistically, Pearson correlation coefficient is calculated between the perioperative SampEn and BIS for both the filtered signal and raw signal, as displayed in Fig. 4. The coefficient displayed a significant increase from 0.51 ± 0.17 to 0.72 ± 0.09 (p < 0.05). Thus, MEMD filtering improved the purity of a signal although we only used SampEn as an example to demonstrate this filtering effect.

Figure 3 One representative case of demonstrating filtering effect using N-A-MEMD.

(A) BIS value, (B) SampEn from raw EEG data and (C) SampEn from filtered data. Clearly, the raw data SampEn fluctuates more sharply than the filtered one (C). The pattern behaves more similarly between (A) and (C). SampEn, sample entropy.

Figure 4 Correlation coefficient between SampEn and BIS for both raw EEG data and filtered EEG data.

Statistics demonstrate the significantly increased filtered capability of empirical mode decomposition. Asterisk *, significant difference (p < 0.05).

Comparison with recurrence analysis and permutation entropy

Discrimination of the different anaesthetic stages

The awake stage denotes the period before the anaesthetic injection. Patients are notably aware during this stage. Then LoC is slowly induced by drugs, eventually followed by a stable unconsciousness state. By the end of the operation, the patients gradually recover consciousness the as the drug volume decreases. Therefore, it is worthwhile to distinguish the stages prior to tracking DoA for the entire surgery. A 2-min noise-free segment was extracted from the three stages for each patient. By applying a window size of 30 s and a window step of 5 s, 24 data points were calculated for each stage, and then an average of the points of all cases were taken. Figure 5 displayed four indexes percentile distribution of the three stages—RQA-RR, RQA-DET, PEn, and SampEn. All the indexes percentile distributions displayed a significant difference between the unconsciousness and conscious state (p < 0.05). Among them (Figs. 5A and 5D), the RR and SampEn were more robust and compatible to the awake and recovery period. Therefore, these distributions were more capable of processing the consciousness stage data and reducing the risk of classifying the EEG into a wrong stage. Moreover, both distributions were more sensitive to the recovery from anaesthesia. Both RQA results increased during anaesthesia, whereas the entropy ones had a low value at LoC (Figs. 5C and 5D). This finding is consistent with the stable and less complex brain activities during unconsciousness.

Figure 5 Comparisons between SampEn and the other three indexes in separate anaesthetic stages.

Boxplots of recurrence analysis measurements: recurrence rate (A) and determinism (B), and Entropy measurements: Permutation Entropy (C) and Sample Entropy (D) for all dataset cases are presented respectively. From the results, it can be easily observed that all four indexes can distinguish the unconscious state. Entropy indexes drop during LoC, while the recurrence analysis measurements rise. Note that the SampEn and RR can accurately measure the similar value between awake and recovery stage without statistical difference. Loc, Loss of consciousness; RR, recurrence rate; DET, determinism; PEn, Permutation Entropy; SampEn, Sample Entropy. Significant difference asterisk: * p < 0.05 and ** p < 0.01; n.s, not significant.

Perioperative performance of evaluating DoA

To find the optimal index to symbolise DoA, a further statistical analysis was conducted. By regarding the corresponding perioperative BIS value as the gold standard, the correlation coefficient, MAE, and AUC under ROC were obtained, as listed in Table 2. From the comparisons, we observed that the entropy method had a much better capability than the RQA group. Moreover, SampEn had the highest correlation with BIS (0.72 ± 0.09), and the corresponding AUC was the highest as well (0.95 ± 0.04). The AUC usually ranged from 0.5 to 1. The more effectively a method classified input, the larger the value was. This provided strong evidence for using SampEn to assess DoA in this study.

Table 2 Comparisons between SampEn and other typical nonstationary and nonlinear analysis methods.

It can be seen that the RQA analysis generally presents negative correlation with BIS although DET has relatively higher correlation with BIS. Compared to entropy methods, the AUC and MAE show inferior values. Both SampEn and PEn have high AUC values, however, the SampEn shows better correlation.

	CC	AUC	MAE	
RR (RQA)	−0.16 ± 0.16	0.24 ± 0.13	48.60 ± 7.95	
DET (RQA)	−0.39 ± 0.16	0.16 ± 0.10	58.33 ± 7.98	
PEn	0.55 ± 0.24	0.91 ± 0.09	42.81 ± 7.92	
SampEn	0.72 ± 0.09	0.95 ± 0.04	47.23 ± 7.84	
Notes.

MAE is not an absolute metric because it can decrease through regression. Values are presented with mean ± STD. Markers symbolize the statistical significant Student t-test.

PEn Permutation

CC correlation coefficient

AUC area under curve

MAE mean absolute error

RQA recurrence quantification analysis

RR recurrence rate

DET determinism

SampEn sample entropy

Regression model performance

To improve the capability of the selected SampEn index for predicting DoA, non-linear regression models including SVM, ANN and Random Forest, were used. Table 3 provides statistical result of output performance of the three models. It can be clearly confirmed that Random Forest generated the best values in terms of the correlation coefficient (0.90 ± 0.05), AUC (0.98 ± 0.02) and MAE (5.22 ± 1.12) among all three models. In addition, it is found that SVM had a statistical result that is similar to ANN, which implied their approximate effect on the data. Moreover, this can be observed in Fig. 6, in which the Random Forest (Fig. 6C) showed a strong linear relationship with BIS, of which the R2 is the highest. Clearly, Figs. 6A and 6B displayed a similar scatter distribution, which was consistent with the results displayed in the Table 3.

Table 3 Comparisons of performance of the three different regression methods.

Obviously, the Random Forest demonstrates the best capability to correlate the SampEn with BIS. SVM and ANN exhibit similar regression effects. For Random Forest, it has the best performance in terms of all the metrics among all models.

	CC	AUC	MAE	
SVM	0.76 ± 0.09	0.95 ± 0.04	9.08 ± 2.72	
ANN	0.79 ± 0.08	0.96 ± 0.04	8.88 ± 2.46	
Random Forest	0.90 ± 0.05	0.98 ± 0.02	5.22 ± 1.12	
Notes.

Values are presented with mean ± STD. Markers symbolize the statistical significant Student t-test.

SVM support vector machine

ANN artificial neural network

CC correlation coefficient

AUC area under curve

MAE mean absolute error

Figure 6 Scatter plots between BIS values and three model outputs.

A total of 33,895 data pairs come from 24 patients: (A) SVM output vs BIS; (B) ANN output vs BIS; (C) Random Forest output vs BIS. Generally, SVM and ANN present similar distribution. Both have close R2 values. Notably, the Random Forest (C) performs best. The output shows a strong linear relationship with the BIS gold standard. SVM, support vector machine; ANN, artificial neural network; BIS, bispectral index.

ANOVA analysis of four data-pair subgroups

By considering BIS as the gold standard, four groups were obtained–Phase 1 (0–40, deep anaesthesia), Phase 2 (40–65, general anaesthesia), Phase 3 (65–85, light anaesthesia), and Phase 4 (85–100, awake). The corresponding model output was automatically divided into these phases. Thus, 33,895 data pairs (i.e., models’ output and BIS) existed for the whole procedure over 24 patients. Statistical analysis was performed using one-way ANOVA on ranks and Student-Newman-Keuls test for pairwise comparisons. The results showed a general ascending trend and a significant difference when compared with its previous subgroups (p < 0.05) for each model, as shown in Fig. 7. The variation of Random Forest (Fig. 7C) in each stage was the smallest, and the gap between different phases was the largest; this implied a superior DoA tracking ability. Moreover, in Tables 4–6, the values distribution in each phase was presented. For all the phase groups, the majority values fell into the corresponding phase definition range except Phase 4 in SVM and Phase 3 in ANN (detailed in the captions of Tables 4–6). This behavior can be more clearly observed in Figs. 6A and 6B of the scatter plot. Note that the Random Forest had the optimal subgroup value distribution, which was consistent with Fig. 6C. This demonstrated a general linear relationship between the two variables. The Random Forest regression performed the best regression effect.

Figure 7 Comparisons of performance of the three different kinds of regression models.

Different BIS phase’ values and corresponding SampEn regression outputs derived from three learning models: (A) SVM; (B) ANN; (C) Random Forest. Even though different phases in each model can be statistically different from each other, the mean gap between every two nearby phases in (C) is much larger and the variation in each phase is much smaller than corresponding phase in SVM (A) and ANN (B). It demonstrates the good fitting ability of Random Forest among these methods. The results are consistent with Tables 4–6. Data are presented as mean ± SD; SampEn values obtained during the different phases were compared by one-way ANOVA on ranks and Student-Newman-Keuls test for pairwise comparisons. The asterisk (*) indicates that SampEn values in each phase are significantly different with P < 0.05 when compared to the values of the preceding phase.

Table 4 SVM outputs distribution across each phase.

All model outputs mostly fall into the corresponding phase definition ranges (e.g., in Phase 1: BIS 0–40 row, the largest output part (73.0%) belongs to 0–40 range bin) except phase 4: BIS 85–100 row, during which the top one falls into 65–85 bin.

BIS Level		Model output	
	n	0-40	40-65	65-85	85-100	
BIS 0–40	15,467	11,293 (73.0%)	4,090 (26.4%)	84 (0.6%)	0 (0%)	
BIS 40–65	13,952	6,302 (45.2%)	7,348 (52.7%)	287 (2.1%)	15 (0%)	
BIS 65–85	2,274	83 (3.6%)	899 (39.5%)	1,008 (44.4%)	284 (12.5%)	
BIS 85–100	2,202	47 (2.1%)	508 (23.1%)	872 (39.6%)	775 (35.2%)	

Table 5 ANN outputs distribution across each phase.

All model outputs mostly fall into the corresponding phase definition ranges (e.g., in Phase 1: BIS 0–40 row, the output largest part (59.3%) belongs to 0–40 range bin) except phase 3: BIS 65–85 row, during which the top one falls into 40–65 bin.

BIS level	Model output	
	n	0–40	40–65	65–85	85–100	
BIS 0–40	15,467	9171 (59.3%)	6,219 (40.2%)	76 (0.5%)	1 (0%)	
BIS 40–65	13,952	4218 (30.2%)	9,466 (67.8%)	248 (1.7%)	20 (0.1%)	
BIS 65–85	2,274	47 (2.1%)	958 (42.1%)	851 (37.4%)	418 (18.4%)	
BIS 85–100	2,202	24 (1.1%)	547 (24.8%)	623 (28.3%)	1008 (45.8%)	

Table 6 Random Forest outputs distribution across each phase.

All model outputs dominantly fall into the phase definition value ranges (e.g., in Phase 1: BIS 0–40 row, the output largest part (82.9%) belongs to 0–40 range bin), which shows a strong linear relationship with BIS. This is consistent with Fig. 6C. It proves the superiority in subgroup classification.

BIS level		Model output	
	n	0–40	40–65	65–85	85–100	
BIS 0–40	15,467	12,820 (82.9%)	2,647 (17.1%)	0 (0%)	0 (0%)	
BIS 40–65	13,952	2,622 (18.8%)	11,091 (79.5%)	239 (1.7%)	0 (0%)	
BIS 65–85	2,274	0 (0%)	761 (33.5%)	1,349 (59.3%)	164 (7.2%)	
BIS 85–100	2,202	0 (0%)	13 (0.6%)	836 (38.0%)	1352 (61.4%)	

Discussion

In this paper, a novel method for estimating DoA was proposed. By extracting EEG signal features, the anaesthesia-related EEG patterns were acquired with different anaesthetic levels. After selecting the SampEn as the input, three regression models were established and compared in terms of the regression effect by taking BIS as a DoA reference. The distribution of the proposed output index in four different subgroups (phases) demonstrated a linear relationship with BIS, of which the Random Forest performed the best. Results illustrated the capability of the proposed method to signify DoA over 24 patients. The analysis is focused on the perspective of the time domain features, which could facilitate DoA research method exploration. From the time domain, instead of only the spectral angle, this methodology might provide a universal solution to those situations where the commercial index is not applicable, such as BIS deficiency in ketamine or nitrous oxide (Avidan et al., 2008).

EEG serves a vital role for brain-related disease studies such as epilepsy, and Parkinson’s diseases due to its strong connection with brain activity (Le Van Quyen et al., 2001; Ly et al., 2016; Vlisides & Mashour, 2017)). Similar to anaesthesia, EEG traces show dominant low-frequency and high-amplitude oscillations. However, the high frequency components decrease from common medication, and thus provide the opportunity to develop EEG-based methods to track DoA automatically (Purdon et al., 2015). However, EEG is easy to be contaminated by EOG, EMG baseline drift and nonlinear distortion of the amplitude and so on (Liu et al., 2017a). Moreover, due to its non-stationary and non-linear properties, the traditional filtering method may not work appropriately. In our procedure, MEMD produced an optimal artefact cancellation capability. By comparing the correlation coefficients in Fig. 4, it is observed that the filtered effect is significantly improved from 0.51 ± 0.17 to 0.72  ± 0.09 (p < 0.05). Although MEMD is an improved EMD method (Rehman & Mandic, 2009), it still requires a long computation time. Therefore, MEMD cannot be applied to real-time situations currently. This drawback of MEMD might be overcome due to hardware development. Another point of concern is the principle of selecting the combination of IMFs (Huang, 2014; Komaty et al., 2014). The decomposition result varies on an individual basis because EEG signals might vary from patient to patient, although IMF2 + IMF3 were selected based on previous study (Huang et al., 2013). Further exploration might reveal superior combinations. These properties should be analysed more precisely, in terms of different anaesthetics, age group, and so on. Another solution is that ICA (Long & Sheng, 2014), or wavelet (Mamun, Al-Kadi & Marufuzzaman, 2013) can be employed for obtaining superior artefact rejection (Khatwani & Tiwari, 2013).

Entropy is an optimal concept for the series randomness analysis (Fast, 1962), and is becoming a powerful analysis tool for information process of the EEG activity currently (Al-Kadi, Reaz & Ali, 2013; Chu et al., 2017). Entropy implies the degree of irregularity of a system. Therefore, we observed significant decrease in the entropy values during anaesthesia, as shown in Fig. 4. However, the factors are chosen empirically. It is worthwhile to study the rationale of proper factors to improve the measurement accuracy further. Results should be compared with a range of these parameters. Moreover, an EEG signal naturally exhibits complicated variations, which occur from complex self-regulating systems over multiple temporal scales. Therefore, the advanced multiscale entropy family, such as multiscale SampEn or multiscale PEn, can be considered as alternative methods although PEn was inferior to SampEn in our data analysis (Shalbaf et al., 2013). Moreover, brains spatial difference suggest that multichannel analysis should be employed, which may present a different level relationship related to anaesthesia.

As observed in the regression step, Random Forest proves to be best. The correlation coefficient, AUC, and MAE of the Random Forest were the highest compared to those of the other two models. SVM and ANN demonstrated similar capability of regression. Both models largely addressed the gap between SampEn and BIS; however, their correlation coefficient and AUC remained approximate. Random Forest improved the metrics significantly. Because we chose the typical and most widely used parameters for each model based on literature (Chang & Lin, 2011; Li, Cui & Voss, 2008; Liu et al., 2016; Ripley & Ripley, 2001; Shalbaf, Behnam & Moghadam, 2015), more research on the appropriate parameters may be required. Moreover, these learning models rely on large datasets to improve themselves. Therefore, a higher amount of data should be provided for training. Moreover, some other machine learning methods, such as deep neural network or boosting, could be applied to increase the accuracy of DoA assessment.

Our present work has several limitations. First, the BIS value has some time delay (Shalbaf et al., 2013) and drawbacks for paediatric application somehow. Therefore, the Expert Assessment of Consciousness Level, which is obtained from experienced anaesthesiologists’ comprehensive evaluation through considering multiple clinical indicators used in practice, would be referred to be as the gold standard for our method aimed at minimising drawbacks and maximising universality. Second, the BIS has been a subject of some controversy on the measurement of DoA over different drugs (Avidan et al., 2008), because the BIS may not correctly reflect patients status when NMDA-dependent anaesthetic agents such as ketamine and nitrous oxide are used; NMDA-dependent anaesthetic agents induce anaesthesia by enhancing inhibitory receptors. This fact explains why BIS (i.e., when using EEG) is not considered as a compulsory standard in practical operation. However, the BIS correctly reflects patients status when GABA-dependent anaesthetic agents such as propofol and desflurane, which induce anaesthesia by suppressing the excitatory nervous circuits, are employed (Uhrig, Dehaene & Jarraya, 2014). Therefore, 24 patients received the GABA receptor dependent drugs in this study. BIS could be selected as the gold standard for our methodology. However, this implies that our method might not be applied to other agent with different anaesthesia mechanisms such as ketamine, nitrous oxide mentioned previously. This needs to be confirmed whether it works as well taking new gold standard instead of BIS, which could exemplify rationale of our methodology. Furthermore, the sample size could be improved further in order to decrease the sample variations, although 24 cases are initially required to obtain statistical results. Fourth, only adults were selected for our study. There is a considerable difference between adults and infants from the clinical viewpoint (Schechter, Allen & Hanson, 1986). Thus, our method may not be applicable to children. Fifth, our definition of the four groups of different anaesthetic levels may be questionable. Further appropriate categorisation of the groups might be required. Analysis with respect to this should be more comprehensive. Finally, the estimation of DoA with a single parameter such as the EEG measure does not work well over time. Accurate DoA monitoring might be improved by taking activity of the autonomic nervous system into consideration (Cohen, Cameron & Duncan, 1990). Therefore, even though numerous trials have sought to integrate several modes of observations for increasing the accuracy of DoA evaluation, there is no single perfect index. Overall, these challenges provide many possibilities for researches to explore a more advanced DoA assessment in future.

Conclusions

In this paper, a novel method protocol based on the SampEn concept and Random Forest learning method for estimating DoA through EEG signals was proposed. The method provides optimal performance compared with other typical methods. First, the SampEn feature derived from EEG signals distinguished the three selected traditional stages of anaesthesia. Second, the feasibility of MEMD to filter EEG data and regression model effect was demonstrated. By considering the BIS as the gold standard, optimal DoA assessment performance for the entire perioperative duration was verified with ROC. More importantly, ANOVA analysis was conducted for 24 patients induced with GABA dependent anaesthetic agents under different anaesthetic levels, which demonstrates a clear linear relationship with the gold standard. The results exhibited high accuracy of DoA estimation. This work provides a basis for DoA evaluation that may be especially helpful for optimising anaesthetic titration to avoid unwarranted effects when deep anaesthesia is performed inappropriately and for promoting fast postoperative recovery.

Additional Information and Declarations

Competing Interests

Author Contributions

Human Ethics

Data Availability

The authors declare there are no competing interests.

Quan Liu analyzed the data, prepared figures and/or tables, approved the final draft.

Li Ma performed the experiments, analyzed the data, prepared figures and/or tables, approved the final draft.

Shou-Zen Fan performed the experiments, contributed reagents/materials/analysis tools, approved the final draft.

Maysam F. Abbod authored or reviewed drafts of the paper, approved the final draft.

Jiann-Shing Shieh conceived and designed the experiments, prepared figures and/or tables, authored or reviewed drafts of the paper, approved the final draft.

The following information was supplied relating to ethical approvals (i.e., approving body and any reference numbers):

All studies were approved by the Research Ethics Committee, National Taiwan University Hospital (NTUH), Taiwan, and written informed consent was obtained from patients (No: 201302078RINC).

The following information was supplied regarding data availability:

Ma, Li (2017): EEG and BIS raw data. figshare. Fileset. https://doi.org/10.6084/m9.figshare.5589841.v1.

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
