# Peer review of "Sample entropy analysis for the estimating depth of anaesthesia through human EEG signal at different levels of unconsciousness during surgeries"

_PeerJ, doi:10.7717/peerj.4817_

## Round 0.1 · original submission · Major Revisions

First, I apologize for the length of time it has taken for your manuscript to go through the review process - a number of challenges led to numerous delays.

Moving forward, there are a significant number of questions regarding this work from both reviewers. While novelty is not a criteria for publication, it should be made much clearer where this work contributes and how this approach compares to other existing methods, so please fully address the questions and concerns of Reviewer 2.

In addition to the reviewer comments, please consider the following:

- More detail in the clinical data would be helpful - e.g., what was the breakdown of patients having a given type of drug administered?

- Comparison to other methods would be very helpful in interpreting the results. Figures 3-6 essentially just provide a comparison of correlation coefficients between time series. While informative, it would be better to understand this approach in the context of other methods.

- Figure 9 is not explained well.

- Nearly all the figure captions could be improved by providing additional information within them so as to help interpretation of the corresponding figures.

- Strengths and weaknesses of the approach and the study - indicate if there are potential issues with regard to the number of samples in this study and associated consequences, if any.

Finally, the manuscript suffers from a significant number of grammatical and language issues. I have attached a partial set of potential changes that would be a start in improving readability. These suggestions are not comprehensive and I would strongly recommend having a professional service help with edits before resubmitting.

Reviewer 1 ·

Basic reporting

This study is to address a novel method of EEG singals for monitoring the depth of anesthesia. The manuscript is well rewriten, and stucture of this manuscript is good.

Experimental design

Data and analysis is OK, the research issue is definition.

Validity of the findings

The findings of this study are OK.

Additional comments

I have carefully read this manscript. Authors used some existing method to analyze the EEG, including EMD and sample entropy, then used a ANN to model the sample entropy with BIS. I did not find any novel idea and method in this MS, and the sample size is small, too.

Reviewer 2 ·

Basic reporting

This paper studied a new method to assess the depth of anesthesia (DoA). Authors claimed that the SampEn feature derived from EEG is proved to be able to distinguish the selective traditional 3 stages of anesthesia.
The reviewer thought the development of methods/algorithms for automatic classification of EEG is very important because of the large variability of the responses of patients to drugs and noxious stimuli during a surgery. Totally, the abstract is concise and clear for introducing the body of paper, the results sound interesting and conclusion was convincing. The manuscript is well organized and the logic is clear. I only have a couple of concerns I would like the authors to address before publication:

Experimental design

1- The data is not relatively comprehensive. The database should be described in more detail. Please mention type of drugs used during different stage of anesthesia.
2- Introduction should provide more information regarding advantages and disadvantages of proposed method in comparison with the nonlinear measure with an application of depth of anesthesia. Actually, the introduction should provides sufficient research background about current studies of DoA system.
3- Additional information should be provided about the rationality that supports the selection signal processing techniques. What kind of properties can be evaluated by applying them? What is the justification to select the SampEn instead of the frequency-based band or other non-linear features?
Describing the properties of the algorithms will help to understand the properties that must be present in the EEG signal that are related (correlation) to level of consciousness. A better approach is to describe the behavior of the EEG as a function of the depth of consciousness and apply the signal processing algorithms that are suitable to extract the features or the feature values.
4- Is there any preprocessing for EEG signal? It seem that there are many noise embedded in EEG signal.

Validity of the findings

5- The validation of the proposed method presented in the manuscript is quite limited. Please compare your result with permutation entropy methods or recurrence analysis which have used by many authors in recent years.
6- In this paper, the authors have used ANN classifier. The comparison with existing classification methods, such as SVM and Random Forest, should be considered.
7- Discussion section should provide more information regarding advantages and disadvantages of proposed method in theory in comparison with the linear and other non-linear approach such as Permutation Entropy, Fractal measures and recurrence quantification analysis.
8- Statistical analysis is week and I recommend go through more.

Additional comments

9- A brief summary of the significance of each figure and table in its heading would help readers to understand the figure and tables quickly without having go to the very details.
10- The manuscript has to go also through careful proof-reading as some of the text is hard to understand because of the grammatical problems.

---

## Round 0.2 · Major Revisions

While significant additions have been included to help address reviewer concerns, the text and underlying science is still quite difficult to follow. Broadly, although it appears that a language editor was brought in to help improve the writing, fundamental issues remain. These issues are significant enough that I would recommend using a different language editing service with strength in scientific writing. Note that both the main text as well as the figure captions need improved clarity and overall improved language before being acceptable for publication.

While the underlying data being studied has been explained better, detail is still lacking. E.g., a table describing the treatment procedures for the 24 patients would be extremely helpful. While it appears that all were given propofol, others seem to have gotten fentanyl or other compounds. Time periods of the surgeries also appear to have varied from 30 min to 4 hours. A table describing such relevant properties for the patients used in this study would be highly informative.

---

## Round 0.3 · accepted · Accept

Thank you for taking the effort to improve the writing of the manuscript as well as provide additional details - these have significantly improved the manuscript. Congratulations again.